# Rapid Detection of *Pityophthorus juglandis* (Blackman) (Coleoptera, Curculionidae) with the Loop-Mediated Isothermal Amplification (LAMP) Method

**DOI:** 10.3390/plants10061048

**Published:** 2021-05-22

**Authors:** Domenico Rizzo, Salvatore Moricca, Matteo Bracalini, Alessandra Benigno, Umberto Bernardo, Nicola Luchi, Daniele Da Lio, Francesco Nugnes, Giovanni Cappellini, Chiara Salemi, Santa Olga Cacciola, Tiziana Panzavolta

**Affiliations:** 1Laboratory of Phytopathological Diagnostics and Molecular Biology, Plant Protection Service of Tuscany, Via Ciliegiole 99, 51100 Pistoia, Italy; domenico.rizzo@regione.toscana.it (D.R.); giovanni.cappellini@regione.toscana.it (G.C.); 2Department of Agricultural, Food, Environmental and Forestry Science and Technology (DAGRI), Plant Pathology and Entomology Section, University of Florence, Piazzale delle Cascine 28, 50144 Florence, Italy; matteo.bracalini@unifi.it (M.B.); alessandra.benigno@unifi.it (A.B.); tiziana.panzavolta@unifi.it (T.P.); 3Portici Unit, Institute for Sustainable Plant Protection, National Research Council (IPSP-CNR), P. le Enrico Fermi 1, 80055 Portici, Italy; umberto.bernardo@ipsp.cnr.it (U.B.); francesco.nugnes@ipsp.cnr.it (F.N.); 4Florence Unit, Institute for Sustainable Plant Protection, National Research Council (IPSP-CNR), Via Madonna del Piano 10, 50019 Sesto Fiorentino, Italy; nicola.luchi@ipsp.cnr.it; 5Department of Agricultural, Food and Agro-Environmental Sciences, University of Pisa, Via del Borghetto 80, 56124 Pisa, Italy; daniele.dalio@hotmail.com (D.D.L.); chiarasalemi93@gmail.com (C.S.); 6Department of Agriculture, Food and Environment, University of Catania, 95123 Catania, Italy; olgacacciola@unict.it

**Keywords:** bark beetle, invasive species, molecular identification, thousand canker disease, walnut

## Abstract

The walnut twig beetle *Pityophthorus juglandis* is a phloem-boring bark beetle responsible, in association with the ascomycete *Geosmithia morbida*, for the Thousand Cankers Disease (TCD) of walnut trees. The recent finding of TCD in Europe prompted the development of effective diagnostic protocols for the early detection of members of this insect/fungus complex. Here we report the development of a highly efficient, low-cost, and rapid method for detecting the beetle, or even just its biological traces, from environmental samples: the loop-mediated isothermal amplification (LAMP) assay. The method, designed on the 28S ribosomal RNA gene, showed high specificity and sensitivity, with no cross reactivity to other bark beetles and wood-boring insects. The test was successful even with very small amounts of the target insect’s nucleic acid, with limit values of 0.64 pg/µL and 3.2 pg/µL for WTB adults and frass, respectively. A comparison of the method (both in real time and visual) with conventional PCR did not display significant differences in terms of LoD. This LAMP protocol will enable quick, low-cost, and early detection of *P. juglandis* in areas with new infestations and for phytosanitary inspections at vulnerable sites (e.g., seaports, airports, loading stations, storage facilities, and wood processing companies).

## 1. Introduction

*Pityophthorus juglandis* Blackman (Coleoptera, Curculionidae, Scolytinae), also known as the Walnut Twig Beetle (WTB), native to northern Mexico and the southwestern United States, is the main vector of *Geosmithia morbida* Kolařik (Ascomycota, Hypocreales) (GM), the fungus responsible for the Thousand Cankers Disease (TCD) of walnut trees [1,2]. This disease originated in the western US [3] and damages mainly individuals of the non-native black walnut (*Juglans nigra* L.), widely planted in this part of the country as an ornamental and nut-bearing tree, while the walnut species native to the southwestern US, such as *Juglans major* (Torr.) A. Heller (Arizona walnut), are less susceptible [4]. Over the past 30 years, *G. morbida* and *P. juglandis* have spread pervasively in many areas of the western US and have also been introduced to some eastern states [4]. In the newly-invaded areas, the bark beetle also attacks other walnut species whose susceptibility to the fungal disease varies.

Plants showing TCD symptoms have been detected starting from 2013 in Europe [5,6]. This fungus-insect association was found in northern and central Italy—the only European country where TCD is at present reported—mainly on black walnuts, although it was also reported on an English walnut (*Juglans regia* L.) growing close to an infected black walnut plantation [7]. The English walnut, widespread and widely cultivated in Europe for nut production, is considered less susceptible to the disease. However, GM induced cankers on *J. regia* also in the US. In fact, in a survey on TCD incidence in *J. regia* orchards in California, Yaghmour et al. [8] identified many trees of this species and of its Paradox hybrid rootstock (*J. hindsii* × *J. regia*) with TCD symptoms and WTB activity.

*J. regia* is a globally popular and valued tree crop, considered an important traditional nut tree in Europe. Used for human nutrition since ancient times, this deciduous, medium-sized, mesophytic tree can be found throughout southern and western Europe and is widely cultivated in its south-central and southeastern areas [9,10]. Today it can be frequently found intercropped with the non-native *J. nigra*, whose cultivation for timber production has been strongly encouraged by the European Union in recent decades [10]. This consociation between the two species can lead not only to a wider spread of TCD in Europe, but it also increases the likelihood of the disease jumping onto *J. regia*.

TCD is caused by the combined activity of WTBs and GM. *P. juglandis* is not the only beetle on which GM propagules have been found. Various beetle species (Curculionidae: Cossoninae and Platypodinae) collected from GM-infected plants have also been found to carry the fungus on their body, although their effective ability to transmit the infection to walnut trees is not proven [11]. Currently, WTB is the only beetle whose capability to vehiculate the disease on walnuts has been proven [1]. In addition, WTB behavior is fundamental for disease development because TCD is the outcome of multiple, repeated attacks by the beetle on the walnut’s branches and stems, which cause numerous holes, around which cankers then develop and coalesce [12].

Both *G. morbida* and its vector *P. juglandis* are regulated in Europe as quarantine pests, listed in Annex II part B of the Commission Implementing Regulation (EU) 2019/2072 [13]. Special requirements for the import and movement within the Union territory of plants for planting, as well as of wood from the genera *Juglans* and *Pterocarya* (Juglandaceae), are laid down in Annexes VII and VIII of the same EU Regulation. Furthermore, EU Regulation 2016/2031 specifies the general requirements for surveys of quarantine organisms within EU territory [14]. These regulatory measures prove the strong need to monitor Europe for both the presence of new outbreaks and, at entry points, for the import-export of walnut material (plants for planting and timber) to prevent the possible introduction and subsequent spread of this insect-fungus complex.

Current survey protocols for identifying the various stages of *P. juglandis* are not straightforward. Survey protocols are not simple and are difficult to implement due to the cryptic life stages of the parasite hidden within the trees. Adult morphological identification requires entomological experience, as is the case with many small-sized bark beetles. The morphological identification is even more troublesome for the preimaginal stages, which are indistinguishable from other similar-sized bark-beetle larvae [15]. Many of the issues related to WTB are common to those of other wood-boring beetles of quarantine significance. For example, for some species morphological identification is hampered by the existence of cryptic species [16]. Management programs to counter alien pest invasions are negatively influenced by various factors such as inadequate monitoring protocols, lack of international standards for phytosanitary measures, poor coordination between stakeholders, and public resistance to implementation of control strategies [17,18]. Because of these difficulties, the development and implementation of fast and accurate molecular methods to rapidly detect the insect are urgently needed.

One such new, versatile molecular detection method is loop-mediated isothermal amplification (LAMP) [19,20]. This technique is cost-effective and more rapid than real-time PCR assays [21,22,23]. Molecular assays based on LAMP have been developed to diagnose a range of parasitic infections in both humans and animals (e.g., malaria, leishmaniasis, and cysticercosis) [24]. Recently, several LAMP tests have also been devised for the identification of invasive plant pathogens [25,26,27,28] and insect pests [29,30,31], both for field applications and in the laboratory. The LAMP technique is a robust and suitable technique for in situ application owing to its low infrastructure requirements and minimal operator training [32]. The method also allows detection through biological traces, such as insect frass, esuviae, and saliva of target organisms, as also demonstrated for other non-native, invasive species [33,34,35,36].

This study aimed to develop a diagnostic protocol based on LAMP assays, in both real-time and visual detection, for the early and reproducible diagnosis of the preimaginal stages, frass and adults of *P. juglandis*.

## 2. Results

### 2.1. Nucleic Acid Extraction from Frass and Insects

The mean concentration of DNA extracts from frass and WTB adults was 325.6 ± 32 ng/µL and 148 ± 22.1 ng/µL, respectively. The average A260/280 ratios were 2.02 ± 0.2 and 1.72 ± 0.4 for frass and adult insects, respectively. The quality of DNA extracted from insects (Table 1), verified by the qPCR probe [37], was good, with a mean Cq value of 20.8 ± 2.5. Similarly, the LAMP protocol based on Tomlinson et al. [38] on DNA frass from WTB showed a mean value of Tamp of 16.8 ± 3.2 (min:s).

### 2.2. LAMP Assay Conditions

The optimal reaction mix for the real-time LAMP assay consisted of 10 μL Isothermal Master Mix OptiGene (ISO-001), 0.2 μM of F3/B3, 0.4 μM of LoopF/LoopB, 0.8 μM of FIP/BIP, and 2 μL of template DNA (5 ng/μL) in a final volume of 20 μL. The real-time LAMP reaction was performed at 65 °C for 30 min, followed by an annealing analysis from 65 to 95 °C, ramping up by 0.5 °C/s, which determined the formation of melting curves. The melting peak for WTB samples of frass was 90.5 °C ± 0.5 °C (Figure 1).

The optimal visual LAMP reaction mixture was also 20 μL: 2 μL Isothermal Buffer 10×, 0.6 mM dNTPs, 2 mM MgSO4, 0.15 mM HNB, and 0.2 M Betaine, plus the following final concentrations of the LAMP primers: 0.2 μM for F3/B3, 0.4 μM for LoopF/LoopB, 0.8 μM for FIP/BIP, 0.32 U/μL Bst 3.0, and 2 μL of template DNA (5 ng/μL). The visual LAMP protocol was conducted on the DNA of both WTB and non-targets extracted from frass. The reaction was performed at 65 °C for 30 min, followed by an additional cycle of 80 °C for 2 min.

### 2.3. Diagnostic Sensitivity, Specificity, and Accuracy of the LAMP Assay

Assays on target and non-target samples (Table 2) did not show any non-specific amplification, only WTB producing amplification curves. A unique amplification curve was generated by each WTB sample, regardless of the starting matrix, thus confirming the specificity of the LAMP assay. In the case of the visual LAMP assay, only WTB was detected by the LAMP reaction, while none of the non-target organisms were amplified (Appendix A). For both protocols, diagnostic sensitivity, diagnostic specificity and relative accuracy were equal to 100%. The end-point PCR protocols designed to evaluate and compare the analytical sensibility (LoD) were also assayed on all target and non-target samples, showing a diagnostic specificity of 100%, the same as the LAMP assay developed in this study.

### 2.4. Blind Panel Validation of the Assay

In the blind panel test, only WTB samples (frass and adults) amplified, with a mean Tamp value equal to 12.98 ± 0.24 (min:s) and 8.74 ± 0.16 (min:s) from frass and WTB adults, respectively. The average melting peaks were equal to 90.83 °C ± 0.00 and 90.75 ± 0.09. The non-target insects and frass did not amplify (Appendix A). Specificity, sensitivity, and accuracy of the results were all 100%. The visual LAMP data were comparable to the real-time LAMP data as only the WTB samples amplified, while the nontargets gave no amplification.

### 2.5. Repeatability and Reproducibility of the Diagnostic Methods

Repeatability and reproducibility were estimated only on WTB frass samples and showed very low SD values (Table 1), varying from 0 to 0.15 (mean Tamps equal to 12.97 ± 1.06).

### 2.6. Limit of Detection (LoD) of the LAMP Assay and Comparison with qPCR (Probe) and Conventional PCR (End-Point) Assays

The LoD was obtained for both the real-time LAMP assay and for the visual LAMP. Dilutions from 10 ng/µL to 5.12 fg/µL of insect and artificial frass DNA were amplified in triplicate. The LoD for the real-time LAMP assay was 0.64 pg/µL with a Tamp value of 13.96 ± 1.76 (min:s) for adult insects and 3.2 pg/µL, with a Tamp value of 13.59 ± 1.38 (min: s), for WTB frass. In the visual LAMP assay, the LoD was similar to that of the real-time LAMP assay (Table 3 and Table 4, Figure 2, Figure 3, Figure 4 and Figure 5).

## 3. Discussion

Vectors are fundamental in the epidemiology of many diseases [39,40]. This is especially true for those diseases caused by pathogens that, like *G. morbida*, require an active vector for being successfully introduced into plants. In these pathogen-vector associations, a prompt detection of the insect vector at an early stage of the invasion process constitutes an important breakthrough, especially in surveillance efforts targeting non-native species at points-of-entry and initial outbreaks in uninfested areas. Hence, a quick identification of WTB, allowing for the timely phytosanitary felling of affected plants, could successfully reduce beetle infestations and thus the amount of TCD. However, correctly distinguishing *P. juglandis* is not always straightforward, as this beetle can be mistaken for bark beetles of similar size; furthermore, distinctive diagnostic features for the preimaginal stages are also lacking. Morphological identification can therefore be technically demanding, requiring time and expert personnel. Moreover, when a large number of samples have to be examined, these checks take on a greater weight [15]. Efficient, high-performance molecular diagnostic methods that provide accurate non-morphological identification of WTB would offer new opportunities for effectively managing this EU-regulated pest complex.

We have developed such a species-specific, highly performing LAMP assay for monitoring *P. juglandis*, even in the absence of specimens on attacked trees abandoned by beetles. The method has also proven to be effective with only biological traces, such as frass (average Tamp 12.97 ± 1.06). Moreover, the assay was not affected by the degradation of the initial matrix. In fact, *P. juglandis* DNA was positively detected even from frass collected from dry twigs stored for two years at room temperature.

To analyze the presence of targets, accurate and reliable DNA extraction from sample matrices with different physical and chemical properties are required. Our protocol confirmed the effectiveness of our extraction method [31], which provided good and reproducible results with all the tested matrices. Furthermore, our method is quick; up to 24 samples (single insects or frass) could be processed in about an hour. The overall LAMP protocol took roughly an hour and a half, from sample preparation to nucleic acid extraction and isothermal reaction. Amplificability tests to check the efficacy of the extraction protocols revealed the absence of inhibitors in DNA extracts from either insects or frass. Performance assays to assess the diagnostic accuracy gave values of 100%. The LAMP test was highly specific, as confirmed by the absence of cross-reactions with various non-target species; this result was further corroborated by the 100% correspondence of amplicons to their homologous sequences. Moreover, the internal blind panel test, performed for both real-time and visual LAMP, showed a precise correspondence among the results obtained in the same laboratory by different operators.

In the molecular detection of plant parasites, the determination of analytical sensitivity of the assay is crucial. In our study, analytical sensitivities starting from serial 1:5 dilutions from both WTB adults and insect frass proved high, with limit values of 0.64 pg/µL and 3.2 pg/µL, respectively. These figures fall within the ranges obtained in similar investigations [31,35,36]. A comparison of the method (both real-time and visual) with conventional PCR did not reveal significant differences in terms of LoD, in contrast to similar investigations targeting other organisms [35]. This was probably due to a sound design of end-point PCR primers optimized in SYBR Green qPCR (data not shown). Conversely, when compared with a qPCR assay that used a hydrolysis probe [35], our new LAMP protocols showed lower analytical sensitivity in assays carried out only with adult insects: 25.6 fg/µL (qPCR) vs. 0.64 pg/µL (LAMP). The same analytical comparison could not be performed using *P. juglandis* frass due to the unavailability of this biological sample in previous investigations [31]. The repetition of the qPCR Probe with the same samples analyzed in LAMP showed, conversely, a better analytical sensitivity of the LAMP compared to the first technique. Finally, repeatability and reproducibility proved excellent, with standard deviation values of inter-run and intra-run variability lower or equal to 0.5 [41].

Our LAMP assays represent valuable tools for quickly gaging the presence of the *P. juglandis* beetle/GM pathogen complex, which is moving eastward in North America and into Eurasia [12,42,43]. On the European and Asian continents this disease complex, in addition to devastating plantations of the widespread *J. nigra*, could cause the decline of *J. regia* throughout its vast native range, both in urban and orchard landscapes. To prevent the destruction that this complex has wrought in the United States, it is essential that, in the recent-introduction areas (Italy), as well as in those that are currently disease-free (the rest of Europe and Asia), fast, accurate and repeatable diagnostic tools become available for prompt detection of early outbreaks. These methods will allow surveying for TCD at regular intervals in quarantined areas and buffer zones, as well as checking plants for planting and fresh wood with bark (i.e., timber, logs, and firewood).

Several DNA amplification technologies have been set up for detecting the members of this insect-fungus complex [31,44,45,46]. We have developed a cheap isothermal amplification method that guarantees simplicity, sensitivity, and reproducibility equal or superior to previous methods. Moreover, it has the great advantage of being portable, so it can easily be used either at entry points (e.g., ports, airports) or at more general inspection sites (e.g., nurseries, loading stations, storage facilities, and wood processing companies). Given these promising results, we are working on the development of a similar protocol focused on the fungal partner (GM).

## 4. Materials and Methods

### 4.1. Sampling

The investigation was performed on a *J. nigra* plantation (43°46′ N, 11°25′ E, about 115 m above sea level) in the province of Florence (Tuscany, Italy), where an outbreak of this insect/fungal disease complex had previously occurred [42,43]. In April 2018, 24 symptomatic samples (small branches, about 26 cm in diameter) were randomly collected from walnut trees throughout the plantation. Branches were checked for the presence of WTB emergence holes on their surface, then bark was peeled off to check for underbark tunnels (Figure 6); adult insects were sampled from galleries and stored in 70% ethanol. The beetle had been previously molecularly identified by end-point PCR amplification [31,42] by targeting a portion of the mitochondrial cytochrome oxidase subunit 1 (COI) gene [47]. All the collected twig samples were stored at room temperature for about a year and a half before being further processed. At this time, the phloematic tissue harboring frass was collected from the twigs and used for the tests. To validate the diagnostic method developed in the present study, different non-target xylophagous insects were also LAMP tested (Table 2). These included insects associated with the same host plants and/or species taxonomically related to the target *P. juglandis*, whose early developmental stages can be mistaken for those of WTB. Adults of all species were included in the assay except for *Zeuzera pyrina* (Linnaeus) (Lepidoptera: Cossidae), for which only the larvae were tested. For both the WTB and *Xylosandrus compactus* (Eichhoff) (Coleoptera, Curculionidae, Scolytinae) the frass was also tested in addition to the adult specimens. Non-WTB adult insects were likewise stored in 70% ethanol, and all insect adults were identified by morphological traits [48]. In some cases, freshly collected material (adult insects, larvae, or frass) was directly used for DNA extraction. In other cases, dried samples, stored at the University of Florence (UF), the Council for Agricultural Research and Economics-Agriculture and Environment (CREA-AE), and the University of Pisa (UP) were used (storage at 12–16 °C, relative humidity: 50%).

### 4.2. DNA Extraction

The DNA extraction protocol was performed both on frass (tested fresh and from samples stored up to two years) and adult insects (Table 2), according to Rizzo et al. [31], with modifications. Collected frass samples (c. 600 mg) were homogenized (15 sec; 30 oscillations/sec) in 10-mL steel jars by using a Mixer Mill MM 200 (Retsch, Torre Boldone, Italy). DNA extraction was performed by the 2% CTAB extraction method and was followed by purification [36] with the Maxwell^®^ RSC PureFood GMO purification kit and authentication kit provided with the automated purificator MaxWell 16 (Promega, Madison, WI, USA), according to the manufacturer’s protocol (Catalog number selected: AS1600). The amount of DNA (ng/μL) and the A260:280 ratios were assessed for each extract using the QIAxpert spectrophotometer (Qiagen, Hilden, Germany). In addition, to verify the quality (and the relative amplificability) of the DNA extracted from the insect and frass samples, a qPCR [37] with a probe (18S Universal rRNA) and LAMP [38] reaction (COX gene) were carried out, using 10 ng in both reactions.

### 4.3. Design of the LAMP and Conventional PCR End-Point Primers

LAMP reactions were carried out using six primers (F3/B3, FIP/BIP, and LoopF/LoopB), designed to specifically target the 28S ribosomal RNA gene of the WTB isolate PR09-635 (accession number: KP201676.1). Primers (Table 5) were designed using the LAMP Designer software (OptiGene Limited, Horsham, UK) and synthesized by Eurofins Genomics (Ebersberg, Germany). A better view of the annealing sites of the LAMP primers on the nucleotide sequence, as well as the amplicon produced, is shown in Figure 7.

The specificity of the primers was further tested using BLAST^®^ (Basic Local Alignment Search Tool; http://www.ncbi.nlm.nih.gov/BLAST, accessed on 24 February 2021) [49]. Moreover, sequences similar to the WTB LAMP amplicon were downloaded from GenBank and used for alignments to emphasize the in-silico specificity of the primers designed in the study. The alignments were performed using the MAFFT software implemented in Geneious 10.2.6 [50], set with the default parameters (Figure 8).

In order to compare the analytical sensibility, specificity, and reliability of the developed Real Time and visual LAMP protocols, a conventional PCR (end-point) assay was also developed to detect WTBs (Table 6). Primers for this assay were designed based on the 28S ribosomal RNA gene (accession number: KP201676.1), using the Oligo Architect ^TM^ Primers and Probe Online software (Sigma–Aldrich, St. Louis, MO, USA) with the following specifications: a 100 to 380 bp product size, a Tm (melting temperature) of 55 to 65 °C, a primer length of 18 to 28 bp, and the absence of a secondary structure whenever possible.

### 4.4. LAMP (Real Time and Visual) and End-Point PCR Assay Optimization

The real-time LAMP reactions were performed using the Isothermal Master Mix (ISO-001) made by OptiGene Limited (Horsham, UK) on a CFX96 thermocycler (Biorad, Berkeley, CA, USA) on the samples listed in Table 2. Each isothermal reaction was performed in duplicate, in a final volume of 20 μL and using 2 μL of target DNA. Negative controls (NTC) were included for each reaction. A melting curve was generated by increasing the temperature from 65 °C to 95 °C with a 10 s interval every 0.5 °C, at the end of the LAMP reaction [51].

The real-time LAMP reactions (amplification and melting curves) were analyzed by using the CFX Maestro 1.0 software (Biorad, Hercules, CA, USA). All real-time LAMP parameters (isothermal amplification time, primer concentration, and annealing temperature (through a thermal gradient)) were evaluated according to Rizzo et al. [36]. PCR products were further analyzed using a QIAxcel Capillary Electrophoresis System (QIAgen, Valencia, CA, USA) with the inclusion of a 25 bp DNA marker. The QIAxcel system uses ScreenGel software that determines the base pair number of each amplicon in individual PCR reactions.

In the visual LAMP protocol, LAMP reactions were carried out in duplicate using the Bst 3.0 DNA polymerase (New England Biolabs, Ipswich, MA, USA) in a total volume of 20 µL. Hydroxynapthol Blue (HNB) was included in the reaction mixture [52] so the color change (from violet to blue) was evaluated at the end of the reaction. Parameters were the same as the real-time LAMP (above). The visual LAMP product reactions were observed by the naked eye under natural light and also photographed using a conventional smartphone camera. If the color changed to light blue the samples were positive, while negative samples remained purple. Moreover, to verify the occurrence of LAMP amplification (as performed for real-time LAMP), the LAMP amplicons were analyzed by the QIAxcel Capillary Electrophoresis System (QIAgen) with the inclusion of a 25 bp DNA marker. All set-up and executions of LAMP reactions were done in a conventional lab bench using designated pipettes and filter tips; imaging analysis was performed in separate rooms. Validation of the conventional PCR assay was carried out by optimizing the primer concentration (0.4–0.6 µM) and annealing temperatures (ranging from 52 °C to 60 °C). The target and non-target samples were the same as those used for the LAMP assay. PCR amplifications were run on a MyCycler thermocycler (Biorad, Hercules, CA, USA); the products were subsequently analyzed by the QIAxcel Capillary Electrophoresis System (QIAgen, Hilden, Germany) with the inclusion of a 25 bp DNA marker.

### 4.5. Performance Characteristics of the LAMP Assay

Sensitivity, specificity, and accuracy of the real-time and visual LAMP assays were evaluated after the optimization of the LAMP protocols using the target DNA samples (Table 2). Samples with a Tamp (time amplification—min:s) value [27,53] greater than 30 (min:s) were considered negative. In the case of the visual LAMP, the diagnostic specificity was verified by naked eye assessment of the color change of the reaction mixture. These parameters are in accordance with the EPPO standards on diagnostics: PM7/98-4 [54]. The end-point PCR protocols, designed to evaluate and compare performance characteristics of the LAMP assay, were also tested on all target and non-target DNA samples (Table 2).

### 4.6. Blind Panel Validation of the Assays

An internal blind panel test was performed on 6 WTB adults, 8 WTB frass samples, the non-WTB specimens (Table 2), and 4 *X. compactus* frass samples. The test was carried out in the laboratory of the Plant Protection Service of Tuscany, Pistoia, Italy, using both the real-time and visual LAMP protocols, as reported above. Additionally, the conventional end-point PCR was tested by the internal blind panel. All DNA samples had been previously diluted to a final concentration of 5 ng/µL. Samples were tested in duplicate; negative controls (No Template Control, NTC) were also included. Based on the blind panel results, the true positives, false negatives, true negatives, and false positives were assessed according to the EPPO requirements outlined by PM7/98-4 [54].

### 4.7. Repeatability and Reproducibility

Repeatability and reproducibility parameters were also assessed in the blind panel on 8 samples of WTB DNA extracted from frass. The intra-run variation (repeatability) and the inter-run variation (reproducibility) were assessed through standard parameters, such as mean, standard deviation, and Tamp (min:s). Eight samples in triplicate, diluted to a final concentration of 5 ng/µL, were tested in two separate series. The mean value and standard deviation were calculated for each sample and for each series of samples, to estimate the repeatability. The reproducibility of each protocol was performed in the same way: by comparing the data of two series of samples by two operators on different dates [55,56].

### 4.8. Limit of Detection (LOD)

The detection limit (LoD) was estimated for each methodology using a 10-fold 1:5 serial dilution (from 10 ng/µL to 2.38 fg/µL). Dilutions were carried out on DNA extracted from adult insects and frass. In the latter case, an artificial frass DNA (100 ng/µL) was obtained by adding the frass of another species (*X. compactus*) to 5 ng/µL of adult WTB DNA to reach a final concentration of 10 ng/µL, adding 2 µL per reaction. A total of 10 dilutions were amplified in triplicate using the real-time and visual LAMP protocols described above.

### 4.9. Comparison between Conventional PCR and qPCR

The same serial dilutions used to determine the analytical sensitivity (LoD) of the LAMP assay were applied to a series of amplifications by end-point PCR and qPCR as a diagnostic and sensitivity comparison with the LAMP assay designed in this study. Table 6 lists the protocols used for these comparisons.

## 5. Conclusions

The developed LAMP assay represents a quick, low-cost diagnostic tool for use during phytosanitary inspections for WTB. It is expected to improve survey programs for *P. juglandis* detection using this new diagnostic strategy, also tracking the WTB through its biological traces, which would reveal the insect’s previous presence. Our LAMP assay thus completes and improves the array of diagnostic protocols currently available to detect the harmful TCD disease, of which the WTB is the only known vector.

## Figures and Tables

**Figure 1 plants-10-01048-f001:**
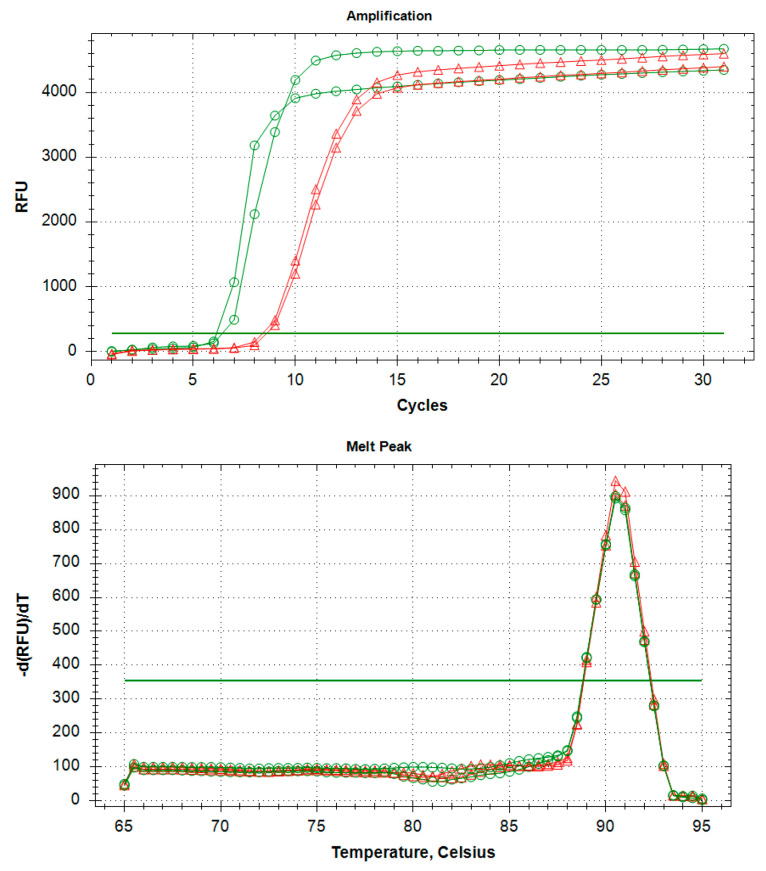
Real-time LAMP amplification curves from adults (green with circles) and frass (red with triangles) of WTB.

**Figure 2 plants-10-01048-f002:**
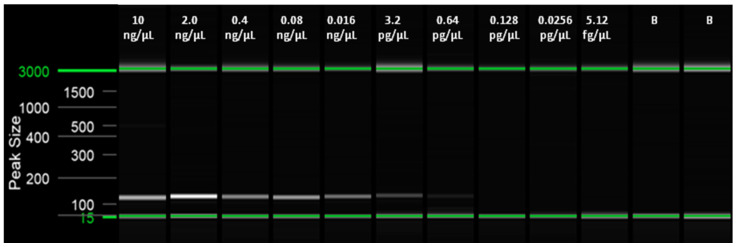
Capillary electrophoresis using the QIAxcel Capillary Electrophoresis System (QIAgen, Valencia, CA, USA) of end point PCR carried out with the primers 14F/125R on serial 1:5 dilutions of WTB adult DNA.

**Figure 3 plants-10-01048-f003:**
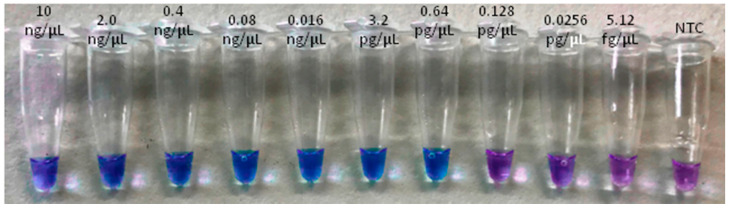
LoD assay of visual LAMP based on WTB adults using serial 1:5 dilutions (from 10 ng/µL to 5.12 fg/µL).

**Figure 4 plants-10-01048-f004:**
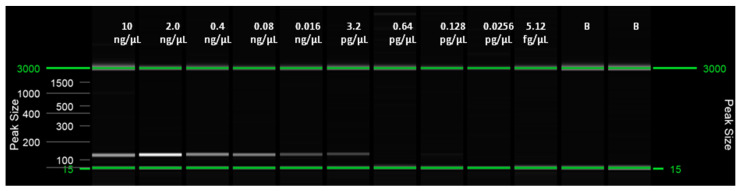
Capillary electrophoresis using the QIAxcel Capillary Electrophoresis System (QIAgen, Valencia, CA, USA) of end point PCR carried out with the primers 14F/125R on serial 1:5 dilutions of artificial DNA from WTB frass.

**Figure 5 plants-10-01048-f005:**
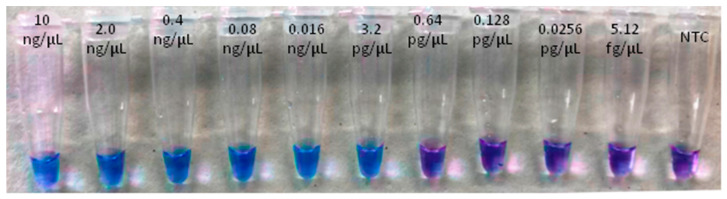
LoD assay of Visual LAMP based on artificial DNA from WTB frass using serial 1:5 dilutions (from 10 ng/µL to 5.12 fg/µL).

**Figure 6 plants-10-01048-f006:**
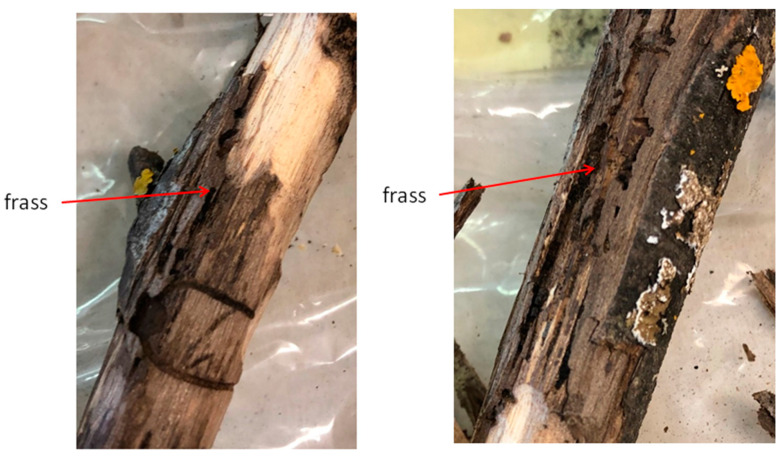
Black walnut branches infested by *P. juglandis* with an indication (red arrows) of the frass sampling sites.

**Figure 7 plants-10-01048-f007:**
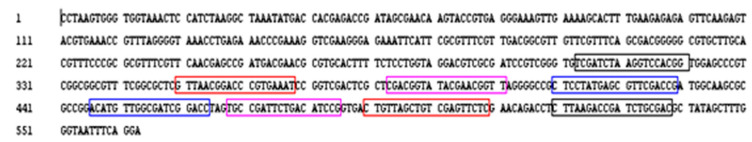
Annealing sites of the LAMP primers on the 28S ribosomal RNA gene sequence of *P. juglandis* (accession number, KP201676.1): F3/B3 (black), LoopB/LoopF (purple), B2/F2 (red), F1c/B1c (blue).

**Figure 8 plants-10-01048-f008:**
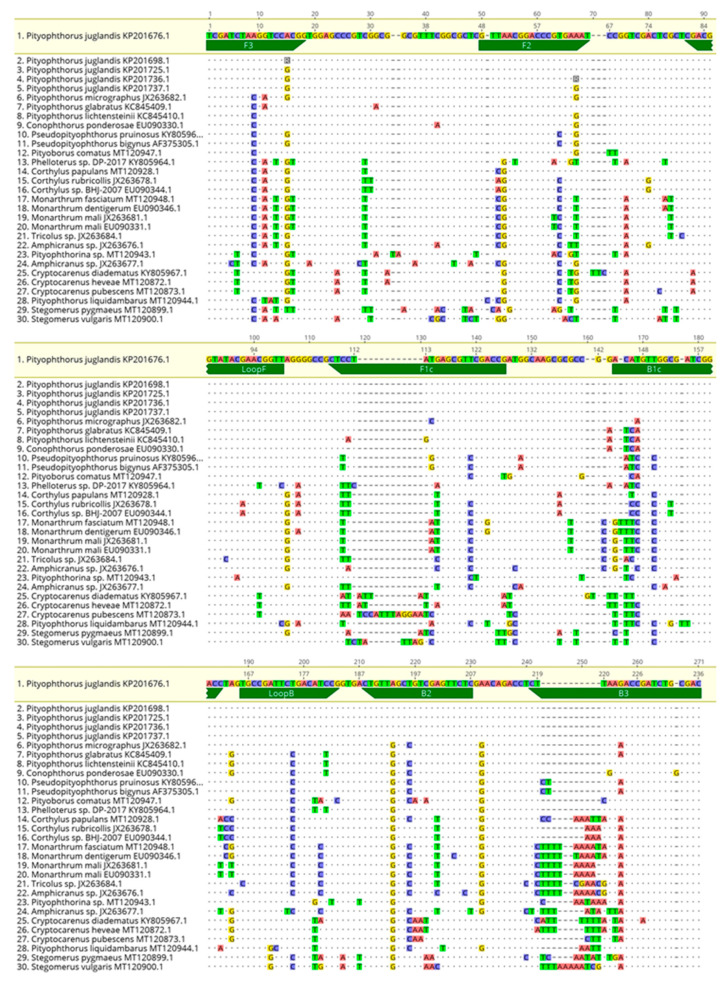
Partial sequence alignment of the 28S ribosomal RNA gene resulting from the in-silico LAMP amplicon of WTB (accession number, KP201676.1) and similar sequences of insects present in GenBank. The alignment is displayed here in three sections to better visualize the differences between the WTB sequence (in yellow) and the homologous sequences.

**Table 1 plants-10-01048-t001:** Repeatability and reproducibility of real time LAMP assay on frass measured as standard deviation (SD).

Sample No.	Real Time LAMP Protocol		
	Repeatability SD		Reproducibility SD
	Assay 1	Assay 2	
1	0.01	0.00	0.04
2	0.05	0.03	0.08
3	0.15	0.01	0.01
4	0.03	0.14	0.06
5	0.04	0.09	0.01
6	0.09	0.06	0.07
7	0.03	0.01	0.03
8	0.04	0.06	0.08

**Table 2 plants-10-01048-t002:** List of target and non-target insects and biological material used in this study. PPS-T: Plant Protection Service of Tuscany, Laboratory of Phytopathological Diagnostics and Molecular Biology; UF: University of Florence; UP: University of Pisa; CREA-AE: Council for Agricultural Research and Economics-Agriculture and Environment (Florence).

Species	Classification	Matrix	Collection Date	Supplier	Source Plant/Device	Hosts
*Pityophthorus juglandis*	Coleoptera, Curculionidae, Scolytinae	frass	2018	PPS-T	*J. nigra*	*Juglans* spp. *Pterocarya* spp.
		adult	2018	UF	*J. nigra*	
*Pityophthorus pubescens* (Marsham)		adult	2018	UF	trap	conifers
*Ips sexdentatus* (Börner)		adult	2018	UF	trap	conifers
*Ips typographus* (Linnaeus)		adult	2014	PPS-T	trap	conifers
*Orthotomicus erosus* (Wollaston)		adult	2018	UF	trap	conifers
*Hylurgus ligniperda* (Fabricius)		adult	2018	UF	trap	conifers
*Tomicus destruens* (Wollaston)		adult	2018	UF	trap	conifers
*Xyleborinus saxesenii* (Ratzeburg)		adult	2018	UF	trap	several host genera (including *Juglans*)
*Anisandrus dispar* (Fabricius)		adult	2020	CREA-AE	*Malus* sp.	several host genera (including *Juglans*)
*Xyleborus monographus* (Fabricius)		adult	2020	CREA-AE	trap	polyphagous (including *Juglans*)
*Xylosandrus compactus* (Eichhoff)		frass	2018	PPS-T	*Laurus nobilis*	several host genera
		adult	2018	PPS-T	*Laurus nobilis*	
*Xylosandrus crassiusculus* (Motschulsky)		adult	2018	UP	*Malus* sp.	several host genera
		adult	2019	UP	*Malus* sp.	
*Xylosandrus germanus* (Blandford)		adult	2019	UF	trap	several host genera (including *Juglans*)
*Lepturges confluens* (Haldeman)	Coleoptera, Cerambicidae	adult	2020	PPS-T	*Juglans* sp.	several host genera (including *Juglans*)
*Zeuzera pyrina* (Linnaeus)	Lepidoptera, Cossidae	larva	2017	PPS-T	*Olea europaea*	several host genera (including *Juglans*)

**Table 3 plants-10-01048-t003:** LoD assay (comparison among different methods) based on WTB adults using 1:5 serial dilutions (from 10 ng/µL to 5.12 fg/µL) and the real-time LAMP protocol. The serial dilutions are the same ones used for the qPCR probe [31] and the Real Time and visual LAMP (this study).

Dilutions	Real Time LAMP		Visual LAMP	qPCR Probe *P. juglandis* (Rizzo et al., 2020a)	End-Point PCR (14F/125R)
	Tamp Means ±SD	Mean Melting Temperatures ±SD	Positive (+)/ -Negative (−)	Cq Means *±* SD	Positive (+)/-Negative (−)
10 ng/µL	6.76 ± 0.42	91.00 ± 0.00	+	18.31 ± 1.15	+
2.0 ng/µL	7.76 ± 0.09	90.75 ± 0.35	+	20.69 ± 0.67	+
0.4 ng/µL	8.25 ± 0.01	90.5 ± 0.00	+	23.05 ± 0.34	+
0.08 ng/µL	9.60 ± 0.54	91.00 ± 0.00	+	24.57 ± 0.21	+
0.016 ng/µL	10.17 ± 0.12	91.00 ± 0.00	+	26.85 ± 0.47	+
3.2 pg/µL	12.92 ± 2.30	90.75 ± 0.35	+	28.35 ± 0.43	+
0.64 pg/µL	13.96 ± 1.76	91.00 ± 0.00	+	30.16 ± 0.17	+/−
0.128 pg/µL	n/a		n/a	32.30 ± 0.05	n/a
0.0256 pg/µL	n/a		n/a	33.53 ± 0.64	n/a
5.12 fg/µL	n/a		n/a	n/a	n/a

**Table 4 plants-10-01048-t004:** LoD assay (comparison among different methods) based on artificial DNA from WTB frass using serial 1:5 dilutions (from 10 ng/µL to 2.38 fg/µL) and the real-time LAMP protocol. The end-point PCR (14F/125R) developed in this study was also evaluated and compared for each dilution.

Dilutions	Real Time LAMP	Visual LAMP	qPCR Probe *P. juglandis* (Rizzo et al., 2020a)	End-Point PCR
	Tamp Means ±SD	Positive (+)/-Negative (−)	Cq Means ±SD	Positive (+)/-Negative (−)
10 ng/µL	6.70 ± 0.01	+	25.67 ± 1.64	+
2.0 ng/µL	7.33 ± 0.02	+	28.69 ± 1.67	+
0.4 ng/µL	8.11 ± 0.06	+	29.05 ± 0.34	+
0.08 ng/µL	9.16 ± 0.08	+	31.64 ± 1.21	+
0.016 ng/µL	9.65 ± 0.59	+	33.76 ± 1.47	+
3.2 pg/µL	13.59 ± 1.38	+	35.64 ± 1.74	+
0.64 pg/µL	n/a	n/a	n/a	n/a
0.128 pg/µL	n/a	n/a	n/a	n/a
0.0256 pg/µL	n/a	n/a	n/a	n/a
5.12 fg/µL	n/a	n/a	n/a	n/a

**Table 5 plants-10-01048-t005:** LAMP primers designed to target WTB DNA. For each primer, the nucleotide position on the reference sequence is reported.

Primer Name	Length (nt)	Sequence 5′-3′	Nucleotide Position	Product Size (bp)	Reference Sequence
Pjug_B3	18	GTCGCAGATCGGTCTTAAG	538-519	160 bp	KP201676
Pjug_BIP(B1c + B2)	39	ACATGTTGGCGATCGGACCGAGAACTCGACAGCTAACAG	446-465		
			509-489		
Pjug_F3	19	TCGATCTAAGGTCCACGG	303-321		
Pjug_FIP(F1c + F2)	39	CGGTCGAACGCTCATAGGAGGTTAACGGACCCGTGAAAT	429-449		
			350-331		
Pjug_LoopB	19	TGCCGATTCTGACATCCG	468-486		
Pjug_LoopF	18	AACCGTTCGTATACCGTCG	401-382		

**Table 6 plants-10-01048-t006:** Conventional PCR and qPCR protocols used for diagnostic comparisons with the LAMP assay. 1 5′-Hexachloro-Fluorescein-CE Phosphoramidite (HEX); 2 Black Hole Quancher 1 (BHQ1).

Primers	Sequence (5′-3′)	Length	Annealing	Type of Protocol	Reference
Pjug_14_F	GCATAGTAGGGACCTCACTTAGTG	112 bp	55 °C	End point	This study
Pjug_125_R	ATAAAGGCATGGGCTGTTACTACA				
Pjug_253_F	TCCCACGTCTTAATAATATAAG	183 bp	55 °C	qPCR Probe	Rizzo et al., 2020a
Pjug_435_R	CTCCTGCTATATGAAGACTA				
Pjug_281_P	Hex_ACTCTTACCACCATCATTAACATTCCT_BHQ1				

## Data Availability

The data presented in this study are available on request from the corresponding author.

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
