# Peer review of "Rapid Detection of *Pityophthorus juglandis* (Blackman) (Coleoptera, Curculionidae) with the Loop-Mediated Isothermal Amplification (LAMP) Method"

_plants, 2021, doi:10.3390/plants10061048_

Round 1
Reviewer 1 Report
The manuscript described the validation of the LAMP assays (Real-time and visual) for the detection of WTB and included the study of the specificity, sensitivity, repeatability and reproducibility parameters. I think the LAMP results were a good work. However, I have some suggestions/remarks as described below. In terms of formatting, please italicize all the species names and re-format all the references according to the journal guidance. The text says that the LAMP primers were designed on the COI sequence, but actually they are designed on the 28S ribosomal RNA gene, please modify the text accordingly.
Title: Capitalize LAMP
Line 20: Abstract: please italicize Pityophthorus juglandis
Line 21: Abstract: please italicize Geosmithia morbida
Line 29 Abstract: please italicize Pityophthorus juglandis
Line 40-41: please rephrase
Line 46 to 49: please can it be rephrased? It is repeating that it has been found in Italy twice
Line 51, specify what is intended by ‘certain degree of susceptibility to the disease’
Line 52 and 53, italicize J. regia
Line 52-53: I think it needs a dot in line 52.
Line 59 to 64: these lines would be better in a new paragraph.
Line 98: italicize in situ
Line 124: modify to: non-targets
Line 132-133: rephrase
Line 144: change respond to amplified.
Line 191, 198, 207, 209, 237,240, 273, 317, 415: Italicize P. juglandis
Line 222: change Visual to visual
Line 243 and 244: italicize J. regia
Line 261: J. nigra
Line 274: Zeuzera pyrina
Line 276: Xylosandrus compactus
Line 278: Please add a reference for the morphological identification.
Line 321: in-silico
Line 401: X. compactus
Other sections
Abstract
- Please add in line 26 that the LAMP assay was designed on the 28S ribosomal RNA gene.
- Include that the LAMP was compared to a qPCR
- Line 26-28: include sensibility levels in pg of DNA/cells (Specify what ‘very small amounts of target insect’s nucleic acid’ means)
Results
2.3. Diagnostic sensitivity, specificity and accuracy of the LAMP assay
- How many target and non-targets were tested? Include this data here
- It needs a table that shows the Tamp means ±SD for all the target samples, it could include the ones in the blind panel validation (It could be supplementary)
2.4 Blind panel validation of the assay:
- Please include also and average of the melting temperature of the WTB samples
2.6. Limit of detection (LoD) of the LAMP assay and comparison with qPCR (Probe) and 154 conventional PCR (end-point) assays
- Table 2 and 3 include the same qPCR Probe juglandis data using WTG adults. Please delete these data from the Table 3. Could these DNA dilutions be checked with the qPCR primers/probe designed by Rizzo et al., 2020a.
- Please include the melting temperature of the real time LAMP products in the Table 2
- I think it is necessary to include the picture of the QIAxcel Capillary Electrophoresis System that confirms the visual LAMP reactions (in Figure 3 and Figure 5).
- Figure 3 caption: change Visual to visual
Discussion
- Please, include in the material and methods the description about the LAMP tests performed with the dry twigs samples (I guess is the material stored at UF but it is not clear that they are the ones samples in 2018 and stored for two years). I would add a table/text that says the average Tamp of these samples.
- Please modify Line 234-235. It says that the LAMP shows greater sensitivity than the qPCR with probe, please rephrase so it is clear that the LAMP detects 0.64pg/ μL and the qPCR 25.6fg/μL
- Line 236-238. Can this new material be checked with the qPCR?
Material and methods:
- Modify Table 7 so it fits in a Line (i.e. Curculionidae in the Classification column and titles do not fit in one line)
- Please, change the 3. Design of the LAMP and conventional PCR end-point primers section, as the primers were designed on the KP201676.1 (28S ribosomal RNA gene)
- Change Figure 7/8 caption accordingly and add the GenBank accession number (modify to 28S ribosomal RNA gene)
- I think the Fig. 8 can be move to Supplementary.
- Line 301, please specify the amount of DNA used in the qPCRs. Was the same amount used in the LAMPs?
- Line 270: how many samples/specie?
- Move lines 348-350 after ‘The real-time LAMP reactions (amplification and melting curves) were analyzed by 344 using the CFX Maestro 1.0 software (Biorad).’
- Line 370. Are more than one sample/specie/matrix? Could you specify how many samples of the target juglaris DNAs samples were tested and used for calculating the diagnostic specificity and sensibility?
Funding section was not completed
Please re-check the bibliography as there are many mistakes in the formatting.

Author Response
Response (in bold) to Reviewer 1
Comments and Suggestions for Authors
The manuscript described the validation of the LAMP assays (Real-time and visual) for the detection of WTB and included the study of the specificity, sensitivity, repeatability and reproducibility parameters. I think the LAMP results were a good work. However, I have some suggestions/remarks as described below. In terms of formatting, please italicize all the species names and re-format all the references according to the journal guidance.
The text says that the LAMP primers were designed on the COI sequence, but actually they are designed on the 28S ribosomal RNA gene, please modify the text accordingly.
The text has been modified accordingly
Title: Capitalize LAMP
Line 20: Abstract: please italicize Pityophthorus juglandis
Done
Line 21: Abstract: please italicize Geosmithia morbida
Done
Line 29 Abstract: please italicize Pityophthorus juglandis
Done
Line 40-41: please rephrase
The sentence has been rephrased
Line 46 to 49: please can it be rephrased? It is repeating that it has been found in Italy twice
The sentence has been rephrased
Line 51, specify what is intended by ‘certain degree of susceptibility to the disease’
The sentence has been made clearer
Line 52 and 53, italicize J. regia
Done
Line 52-53: I think it needs a dot in line 52.
Done
Line 59 to 64: these lines would be better in a new paragraph.
The sentence has been rephrased. In this new version we feel it fits finely in this paragraph.
Line 98: italicize in situ
Done
Line 124: modify to: non-targets
Done
Line 132-133: rephrase
The sentence has been rephrased
Line 144: change respond to amplified.
Done
Line 191, 198, 207, 209, 237,240, 273, 317, 415: Italicize P. juglandis
Done
Line 222: change Visual to visual
Done
Line 243 and 244: italicize J. regia
Done
Line 261: J. nigra
Done
Line 274: Zeuzera pyrina
Done
Line 276: Xylosandrus compactus
Done
Line 278: Please add a reference for the morphological identification.
Done
Line 321: in-silico
Done
Line 401: X. compactus
Done
Other sections
Abstract
Please add in line 26 that the LAMP assay was designed on the 28S ribosomal RNA gene.
Done
Include that the LAMP was compared to a qPCR
Done
Line 26-28: include sensibility levels in pg of DNA/cells (Specify what ‘very small amounts of target insect’s nucleic acid’ means)
Done
Results
2.3. Diagnostic sensitivity, specificity and accuracy of the LAMP assay
How many target and non-targets were tested? Include this data here
Done (Table S1 and S2)
It needs a table that shows the Tamp means ±SD for all the target samples, it could include the ones in the blind panel validation (It could be supplementary)
Done (Table S1 and S2)
2.4 Blind panel validation of the assay:
Please include also an average of the melting temperature of the WTB samples
Done. See also Table S2
2.6. Limit of detection (LoD) of the LAMP assay and comparison with qPCR (Probe) and 154 conventional PCR (end-point) assays
Table 2 and 3 include the same qPCR Probe juglandis data using WTG adults. Please delete these data from the Table 3. Could these DNA dilutions be checked with the qPCR primers/probe designed by Rizzo et al., 2020a.
Done. Table 2 (former ms version) has been eliminated. See integration in the legend of Table 3. Please note that the order of the tables has now changed.
Please include the melting temperature of the real time LAMP products in the Table 2
Done. The melting temperature of the real time LAMP products was included in a column (Table 3)
I think it is necessary to include the picture of the QIAxcel Capillary Electrophoresis System that confirms the visual LAMP reactions (in Figure 3 and Figure 5).
It was not deemed necessary to perform the electrophoretic run because it was not relevant to highlight the result of amplification. This because since the assays were developed in duplicate (both real-time and visual), one test validated the other
Figure 3 caption: change Visual to visual
Done.
Discussion
Please, include in the material and methods the description about the LAMP tests performed with the dry twigs samples (I guess is the material stored at UF but it is not clear that they are the ones samples in 2018 and stored for two years). I would add a table/text that says the average Tamp of these samples.
Done. See S1 and S2
Please modify Line 234-235. It says that the LAMP shows greater sensitivity than the qPCR with probe, please rephrase so it is clear that the LAMP detects 0.64pg/ μL and the qPCR 25.6fg/μL
Done. The sentence has been rephrased
Line 236-238. Can this new material be checked with the qPCR?
The following sentence has been included to comply with this issue: "The repetition of the qPCR probe ...". See also Table 4. - Lod assay (comparison among different methods) ...
Material and methods:
Modify Table 7 so it fits in a Line (i.e. Curculionidae in the Classification column and titles do not fit in one line)
Done
Please, change the 3. Design of the LAMP and conventional PCR end-point primers section, as the primers were designed on the KP201676.1 (28S ribosomal RNA gene)
Done
Change Figure 7/8 caption accordingly and add the GenBank accession number (modify to 28S ribosomal RNA gene)
Done
I think the Fig. 8 can be move to Supplementary.
We deem it more appropriate to keep Fig. 8 in the text as it gives evidence of in silico specificity
Line 301, please specify the amount of DNA used in the qPCRs. Was the same amount used in the LAMPs?
Done
Line 270: how many samples/specie?
Done. See S1 and S2
Move lines 348-350 after ‘The real-time LAMP reactions (amplification and melting curves) were analyzed by 344 using the CFX Maestro 1.0 software (Biorad).’
Done
Line 370. Are more than one sample/specie/matrix? Could you specify how many samples of the target juglaris DNAs samples were tested and used for calculating the diagnostic specificity and sensibility?
See S1 and S2
Funding section was not completed
Done
Please re-check the bibliography as there are many mistakes in the formatting.
Done

Reviewer 2 Report
This article (plants-1183868) by Rizzo and colleagues is well motivated, the structure is appropriate, and the manuscript is well written without missing any key details. The methods used are appropriate for the objectives of the work and, in general, well depicted. The resulting figures are sufficient, informative, and of good quality helping to follow the reasoning throughout the manuscript. The discussion of results and comments on future research was nicely done and will be useful to others. Overall, I enjoyed reading the manuscript. A few editorial remarks have been made below for authors to consider.
L84-90: Some of the authors statements would be much stronger if they tie their work to the body of literature that has built up on palms, ornamental woods, etc. (e.g., I’d suggest adding these papers to the discussion: Milosavljevic et al. 2019 J. Pest Sci. 92: 143-156; Meurisse et al. J. Pest Sci. 92: 13-27). There are many commonalities between the WTB problems addressed in this paper and the similar problems associated with these other quarantine wood boring beetles (i.e., survey protocols are not straightforward and difficult to implement because of pests’ cryptic life stages hidden inside trees; morphological identification troublesome because of cryptic species [e.g., Rhynchophorus ferrugineus vs R. vulneraturs]). Accidental introduction of undetected Rhynchophorus spp.in live palms into new areas has, for example, resulted in establishment and spread of these notorious palm pests in distinctly non-native habitats. Inappropriate surveying techniques, management planning including morphological identifications, poor coordination between stakeholders, and public resistance to implementation of controls continues to adversely affect incursion management programs. Adding this information would benefit the discussion.
Author Response
Response (in bold) to Reviewer 2
Comments and Suggestions for Authors
This article (plants-1183868) by Rizzo and colleagues is well motivated, the structure is appropriate, and the manuscript is well written without missing any key details. The methods used are appropriate for the objectives of the work and, in general, well depicted. The resulting figures are sufficient, informative, and of good quality helping to follow the reasoning throughout the manuscript. The discussion of results and comments on future research was nicely done and will be useful to others. Overall, I enjoyed reading the manuscript. A few editorial remarks have been made below for authors to consider.
L84-90: Some of the authors statements would be much stronger if they tie their work to the body of literature that has built up on palms, ornamental woods, etc. (e.g., I’d suggest adding these papers to the discussion: Milosavljevic et al. 2019 J. Pest Sci. 92: 143-156; Meurisse et al. J. Pest Sci. 92: 13-27). There are many commonalities between the WTB problems addressed in this paper and the similar problems associated with these other quarantine wood boring beetles (i.e., survey protocols are not straightforward and difficult to implement because of pests’ cryptic life stages hidden inside trees; morphological identification troublesome because of cryptic species [e.g., Rhynchophorus ferrugineus vs R. vulneraturs]). Accidental introduction of undetected Rhynchophorus spp.in live palms into new areas has, for example, resulted in establishment and spread of these notorious palm pests in distinctly non-native habitats. Inappropriate surveying techniques, management planning including morphological identifications, poor coordination between stakeholders, and public resistance to implementation of controls continues to adversely affect incursion management programs. Adding this information would benefit the discussion.
Thank you for your valuable comments. We have modified the text accordingly and included the literature you have suggested.
